# Effect of weight loss on the retinochoroidal structural alterations among patients with exogenous obesity

**Aniruddha Agarwal[1], Arshiya Saini[1], Sarakshi Mahajan[2], Rupesh Agrawal[3], Carol Y. Cheung[4], Ashu Rastogi[5], Rajesh Gupta[6], Yu Meng Wang[4], Michael Kwan[4], Vishali Gupta[1] *, for the OCTA Study Group[¶]**

**1** Department of Ophthalmology, Advanced Eye Center, Post Graduate Institute of Medical Education and Research (PGIMER), Chandigarh, India, **2** School of Medicine, St Joseph Mercy Hospital, Oakland, Pontiac, Michigan, United States of America, **3** Department of Ophthalmology, National Healthcare Group Eye Institute, Tan Tock Seng Hospital, Singapore, Singapore, **4** Department of Ophthalmology and Visual Sciences, The Chinese University of Hong Kong, Hong Kong, Hong Kong, **5** Department of Endocrinology, Post Graduate Institute of Medical Education and Research (PGIMER), Chandigarh, India, **6** Department of General Surgery, Post Graduate Institute of Medical Education and Research (PGIMER), Chandigarh, India

¶ Membership of the OCTA study group is provided in the acknowledgments.
* vishalisara@yahoo.co.in, vishalisara@gmail.com

**Data Availability Statement:** All relevant data are within the manuscript and its Supporting Information files.

## Abstract

### Purpose

To evaluate the changes in the retinochoroidal vasculature in patients with exogenous obesity using swept-source optical coherence tomography (SS-OCT) and OCT angiography (OCTA).

### Methods

In this prospective study, 60 patients diagnosed with obesity (47 males) (mean age: 46.47 ±10.9 years) were included, of which 30 patients underwent bariatric surgery (Group A), and 30 patients underwent conservative management (exercise/diet) (Group B). Parameters including choroidal thickness (CT), choroidal vascularity index (CVI) and retinal capillary density index (CDI) and arteriovenous ratio (AVR) were measured at the baseline and three months follow up. 30 eyes (30 age and gender-matched) of normal participants were included for comparison.

### Results

Baseline CT was lower in 60 participants with obesity compared to controls. Compared with normal subjects, subjects with obesity had higher mean CVI (0.66±0.02 versus 0.63±0.04; p<0.01), smaller FAZ area (0.26±0.07 versus 0.45±0.32; p<0.01), higher CDI (superficial plexus: 0.7±0.04 versus 0.68±0.06; p = 0.04, deep plexus: 0.38±0.02 versus 0.35±0.06; p = 0.01), and lower AVR (0.68±0.05 versus 0.70±0.03 versus; p<0.01). At 3-month after intervention, CT showed a significant increase in participants from Group A (329.27±79μm; p<0.01) but not in Group B from baseline. No significant change was noted in CVI or CDI at

**Funding:** The authors received no specific funding for this work.

**Competing interests:** The authors have declared that no competing interests exist.

3-month in either group compared to baseline. AVR significantly increased in Group B (p = 0.03).

## Conclusion

Subclinical changes in retinochoroidal vasculature occurs in participants with exogenous obesity compared to healthy subjects. Surgical intervention (bariatric surgery) may have a favorable outcome on the choroidal thickness in these patients.

## Introduction

Obesity is commonly caused by a combination of excessive food intake, lack of physical activity, and genetic susceptibility though some cases maybe be caused primarily by genetic predisposition, endocrine disorders, medications, or mental disorders, and is measured in terms of body mass index (BMI) [1,2]. Obesity may increase levels of some vasoconstrictor molecules such as endothelin-1 and angiotensin-II [3,4] and has been linked with various ocular diseases including glaucoma, diabetic retinopathy, cataract, and age-related macular degeneration [5–7]. Therefore, obesity may be associated with alterations in the retinal microvasculature and the choroid by a number of mechanisms, which may be the basis for the various ocular diseases.

Assessment of retinal microvasculature includes measuring diameter of retinal arterioles and venules from fundus photographs [8]. The arteriole-to-venule ratio (AVR) reflects preclinical changes in cerebral and coronary microcirculation. The AVR is calculated using the mean vessel calibre of the largest six arteries (central retinal arteriolar equivalent—CRAE) and largest six venules (central retinal venular equivalent—CRVE). In children with obesity, CRAE has been strongly linked to severe cardiovascular risk factors [9]. Similarly, CRAE has been negatively correlated to fat mass indices in children with obesity, reflecting presence of narrow arterioles in these patients affecting the retinal microcirculation [10]. Obesity also results in wider CRVE and lower AVR that correlate with higher leptin levels, indicating systemic metabolic abnormalities [11]. Improved AVR has been found in patients undergone bariatric surgery after 6 months [12] and 9 months of surgery [13].

Choroidal thickness has been found to be affected by physiological variations such as age, gender, refractive status, as well as several local and systemic diseases [14–17]. Literature shows that the choroidal thickness is lower in patients with BMI>25.0 compared to patients with BMI between 18–24 [18]. Recently, availability of swept-source optical coherence tomography (SS-OCT) and optical coherence tomography angiography (SS-OCTA) makes it possible to have precise measurements including choroidal thickness and vascular density indices [19]. The newer indicator of measuring choroidal vascularity index (CVI) indicates the ratio of choroidal vessels to stroma thus allowing the more accurate insight into the vascular alterations associated with various systemic or local diseases [20]. CVI has been reported to decrease in patients with age-related macular degeneration [21] and intraocular tuberculosis [22], while an increase is noted in central serous chorioretinopathy [23].

The alterations caused by obesity, if any, can be quantified by measuring retinal and choroidal thickness, CVI, and AVR. There is a paucity of studies in the literature that have determined the such retinochoroidal alterations in details. Moreover, the effect of correction of obesity by bariatric surgery or non-surgical measures on retinochoroidal vasculature has not been evaluated. The present study aims to describe the quantitative retinochoroidal vascular

alterations associated with obesity at baseline and its correlation following an intervention (bariatric surgery or conservative management).

## Materials and methods

The index study was a prospective observational study that included 60 adult patients of either gender with exogenous obesity who presented to Department of Endocrinology, Postgraduate Institute of Medical Education and Research (PGIMER), Chandigarh. Patients who fulfilled the following inclusion criteria were enrolled in the study: Age 18–70 years, BMI≥27.5 with exogenous obesity, waist circumference of ≥80 cm, patients willing to undergo weight reduction (either medical or surgical measures). The exclusion criteria were patients with endogenous obesity (such as Cushing's syndrome), patients with retinal pathologies that can affect the retinal/choroidal thickness or vascularity such as diabetic retinopathy, macular degeneration, central serous chorioretinopathy, optic atrophy, glaucoma, and uveitis. Patients where media clarity is obscured by the presence of cataract, vitritis or any other such co-existent pathology that does not allow the acquisition of good images were also excluded. In addition, patients with refractive errors more than +3 or -3 D (high hypermetropia and myopia) were also excluded. The study was approved by the Intramural Institute Ethics Committee (IEC) of Postgraduate Institute of Medical Education and Research (PGIMER), Chandigarh, India. Written informed consent was obtained from the patients enrolled in the study. The study adhered to the tenets of the Declaration of Helsinki.

### Baseline evaluation

Detailed history regarding associated co-morbidities including diabetes mellitus, hypertension, obstructive sleep apnea, osteoarthritis was recorded. A detailed treatment history regarding concomitant medications being taken by the patient for several associated co-morbidities was recorded. A general physical examination was done and parameters including height, weight, BMI, waist index, blood pressure were done. The mean arterial pressure (MAP) was also calculated. The best-corrected visual acuity (BCVA) by Snellen's chart (converted to LogMAR units for statistical analysis), intraocular pressure (IOP) measured by non-contact tonometer, slit lamp biomicroscopy for anterior segment and posterior segment examination were performed. Using the MAP and IOP values, the mean ocular perfusion pressure (MOPP) was also calculated as 2/3 of the difference between MAP and IOP.

### Acquisition of images

Thirty-degree color fundus photographs were acquired on the digital fundus camera (DRI Triton, Topcon®) with images focussed on disc and macula. SS-OCT 3D and 5-Line raster scan of the macula and optic disc were done. SS-OCTA (DRI Triton, Topcon®) using 3 × 3 scans of the macula and optic nerve head were performed. The acquisition of the scans were at least twice and better quality images were selected for further analysis. The acquisition of images was performed between 10.00 am to 12.00 noon in our study (common working hours of the Departments of Endocrinology Obesity Clinic and Ophthalmology). This also helps to minimize the effect of diurnal variations in choroidal thickness [16]. The acquired images were analyzed by three independent observers (AA, RA, and VG) and AVR, retinal and choroidal thickness, CVI, retinal thickness and retinal vascularity index were measured.

## Measurement of arteriovenous ratio

The AVR was calculated at the optic nerve head and macula using the fundus photographs. Retinal vascular caliber was measured following a standardized protocol, based on the revised Knudtson–Parr–Hubbard formula, as described in previous publications [24–26]. The CRAE and CRVE values were used to calculate the AVR for all the participants in the study.

## Measurement of retinal and choroidal thickness

Retinal thickness was measured from the inner border of the internal limiting membrane to the outer border of the retinal pigment epithelium (RPE). Choroidal thickness was measured vertically from the outer border of the RPE to the inner border of the sclera. The upper border was marked at the RPE and the lower border area was below the line of light pixels at the choroid-scleral junction. The measurements were performed at the fovea by two independent graders. The average of the two graders' measurements was used for analysis.

## Measurement of CVI

In order to measure the CVI, image binarization was performed for all the scans obtained from patients. Subfoveal scan (central B-scan) was chosen for image analyses. The image was processed on public domain software Image J (National Institutes of Health, Bethesda, USA). Polygon tool was used to select the total choroid area (TCA), which was added in the region of interest (ROI) manager. After converting the image into 8 bit, Niblack auto local thresholding was subsequently applied which gave the mean pixel value with standard deviation for all the points. On the SS-OCT scans, the luminal Area (LA) was highlighted by applying the color threshold. In order to determine the LA within the selected polygon, both the areas in ROI manager were selected and merged by 'AND' operation of Image J. The composite third area was added to the ROI manager. The first area represents the total of the choroid selected, and the third composite area is the vascular or LA. Stromal area (SA), which corresponds to the interstitial or stromal component of the choroid, was obtained by calculating the differences between TCA and LA. The CVI was calculated by dividing LA by TCA.

## Measurement of retinal capillary network

The retinal capillary network was measured in terms of foveal avascular zone (FAZ) area and capillary density index (CDI). For calculation of FAZ, manual delineation was performed on a third-party software (ImageJ). The area was calculated in square millimetres. A circle with a radius of 1.5mm is centered at the subfoveal region. Using the Niblack thresholding and ROI manager, all images were binarized and converted to 8-bits with a mean pixel value and standard deviation of all points. Subsequently, the LA was highlighted within the circle with the brightness set to 0 and 254. The LA was merged with the corresponding threshold area and measured using ROI manager. The CDI was defined as the percentage of capillary density over the stromal area within the 1.5 mm-radius circle at the macula region. CDI was obtained at both, superficial and deep retinal capillary plexus.

The patients were followed-up at 3 months following intervention (conservative management that included diet therapy and exercise, or bariatric surgery). Patients who underwent surgery (n = 30) were categorized into Group A, and patients receiving conservative therapy (n = 30) were categorized to Group B. At follow-up, they underwent ocular examination, fundus photography, SS-OCT and OCTA. All the above parameters were repeated at 3 months.

We included a normative database of 30 participants from our center with no known ocular or systemic disease (mean age: 33.6 ± 8.53 years; 21 males). Similar parameters, i.e. retinal and

choroidal thickness, CVI, FAZ area, CDI and AVR were calculated, and compared to patients with obesity.

## Statistical analysis

We obtained the sample size by referring to the study by Dogan et al [27] considering equal allocation ratio of 1:1, which will provide the desired effect with 95% confidence and 80% power. The statistical comparison of parameters was carried out using non-parametric tests. Multivariate regression modelling were carried out to study the effect of confounders on choroidal thickness. All the analyses were performed using SPSS version 20.0 (IBM Inc.) and the statistical significance evaluated at 5% level.

## Results

### Study participants, demographics and systemic features

The study included a total of 60 patients with 30 patients in each group followed prospectively. The mean age of all patients was $46.47 \pm 10.9$ years (Table 1). There were a total of 47 males in the study. Both the groups matched in their demographic profile at the baseline with no statistically significant difference in any parameter. The co-morbidities of the study participants and the specific treatments are mentioned in Table 1. The mean height, weight, waist circumference, blood pressure (including MAP), BMI, and other laboratory parameters including glycosylated hemoglobin (Hb1Ac) have been summarized in Table 2 (both at baseline and follow-up). At the end of 3 months, BMI was comparable in both the groups ($36.50 \pm 3.93$ versus $39.68 \pm 31.60$; $p = 0.59$). In the normal control participants, the mean height was $164.13 \pm 12.3$ cm, mean weight was $67.2 \pm 10.64$ kg (mean BMI: $24.90 \pm 2.57$) ($p<0.001$ compared to obese patients). The mean waist circumference in this group was $76.07 \pm 3.76$ cm ($p<0.001$ compared to obese patients).

In Group A, the mean BCVA was $0.007 \pm 0.04$ LogMAR units whereas it was $0.001 \pm 0.01$ LogMAR in Group B. There was no statistically significant change in the BCVA at the end of 3

**Table 1. Demographic features of subjects with exogenous obesity included in the study.**

| Parameters | Control Subjects | Patients | | All Patients |
|---|---|---|---|---|
| | | Group A | Group B | |
| | (n = 30) | (n = 30) | (n = 30) | (n = 60) |
| **Age (in years) [Mean ± SD]** | 33.6 ± 8.53 | 42.93 ± 11.38 | 50.00 ± 9.26 | 46.47 ± 10.9 |
| **Gender [No. (%)]** | | | | |
| Male | 21 (70) | 24 (80) | 23 (76.67) | 47 (78.33) |
| Female | 9 (30) | 6 (20) | 7 (23.33) | 13 (21.67) |
| **Co-morbidities [No. (%)]** | | | | |
| Diabetes | - | 10 (33.33) | 17 (56.67) | 27 (45) |
| Hypertension | - | 18 (60) | 7 (23.33) | 25 (41.67) |
| Osteoarthritis | - | 4 (13.33) | 0 (0) | 4 (6.67) |
| Obstructive Sleep Apnea | - | 8 (26.67) | 1 (3.33) | 9 (15) |
| Others | - | 8 (26.67) | 2 (6.67) | 10 (16.67) |
| **Treatment of comorbidities [No. (%)]** | | | | |
| Oral hypoglycemic agents | - | 10 (33.33) | 17 (56.67) | 27 (45) |
| Antihypertensive agents | - | 18 (60) | 7 (23.33) | 25 (41.67) |
| Lipid Lowering Agents | - | 2 (6.67) | 1 (3.33) | 3 (5) |
| Others | - | 6 (20) | 1 (3.33) | 7 (11.67) |

**Table 2. The baseline and follow-up systemic measurements and laboratory parameters of subjects with exogenous obesity included in the study.**

| Parameters (Mean value) ± Standard deviation (SD) | Group A (n = 30) | | | Group B (n = 30) | | |
|---|---|---|---|---|---|---|
| | Baseline | Follow-up | p value* | Baseline | Follow-up | p value* |
| Systolic blood pressure (mm Hg) ± SD | 129.0 ± 7.6 | 127.8 ± 5.7 | 0.08 | 128.3 ± 10.1 | 127.6 ± 9.1 | 0.66 |
| Diastolic blood pressure (mm Hg) ± SD | 84.8 ± 7.2 | 82.6 ± 5.2 | 0.07 | 80.1 ± 7.7 | 80.9 ± 7.7 | 0.33 |
| Mean arterial pressure ± SD | 99.5 ± 6.8 | 97.6 ± 5.1 | **0.03** | 96.2 ± 7.4 | 96.5 ± 7.4 | 0.53 |
| Weight (Kg) ± SD | 119.4 ± 19.1 | 95.0 ± 15.2 | **0.001** | 86.1 ± 10.0 | 83.0 ± 20.7 | 0.46 |
| BMI (Kg/m$^2$) ± SD | 45.7 ± 5.3 | 36.5 ± 4.0 | **0.001** | 32.3 ± 4.3 | 32.1 ± 31.6 | **<0.001** |
| Waist Circumference (cm) ± SD | 105.5 ± 14.6 | 90.9 ± 13.3 | **0.001** | 92.9 ± 7.9 | 95.2 ± 17.7 | **<0.001** |
| Hb1Ac (%)± SD | 6.3 ± 1.4 | 5.5 ± 0.8 | **0.001** | 6.4 ± 1.4 | 6.1 ± 1.1 | **0.02** |

* Wilcoxon-Signed Rank Test.

BMI: Body mass index; Hb1Ac: Glycosylated hemoglobin.

months. In Group A, the MOPP was 55.78 mm Hg at baseline, and 55.01 mm Hg at 3 months (p = 0.07), whereas in Group B, MOPP was 54.4 mm Hg at baseline, and 54.8 mm Hg at 3 months (p = 0.47).

## Comparison of normal participants with obese patients

In comparison to the normal participants (n = 30), patients with obesity (n = 60) did not have statistically significant differences in either the mean retinal or choroidal thickness values at baseline. However, the mean CVI was higher among normal subjects compared to participants with obesity (0.66 ± 0.02 versus 0.63 ± 0.04, respectively; p<0.01). The FAZ area was significantly lower in normal participants compared to patients with obesity (0.26 ± 0.07 versus 0.45 ± 0.32, respectively; p<0.01). CDI was also significantly higher in normal subjects compared to obese patients (superficial plexus: 0.70 ± 0.04 versus 0.67 ± 0.06, respectively; p = 0.04, deep plexus: 0.38 ± 0.02 versus 0.35 ± 0.06, respectively; p = 0.01). AVR was statistically higher in the normal subjects compared to patients with obesity (0.70 ± 0.03 versus 0.68 ± 0.05, respectively; p<0.01).

Table 3 compares the retinochoroidal parameters between normal participants and participants with obesity.

## Baseline image analyses among patients with obesity in both groups

The baseline retinal and choroidal thicknesses in patients from Group A are listed in Table 4. The mean CVI in Group A was 0.63 ± 0.04. Analyses of SS-OCTA among participants from Group A revealed a mean FAZ area of 0.42 ± 0.13 mm$^2$. The mean CDI among participants in Group A was 0.68 ± 0.06 in the superficial plexus and 0.36 ± 0.03 in the deep plexus. Finally, the AVR among the patients in Group A was 0.68 ± 0.06. The baseline and follow-up values of participants in Group B are listed in Table 5. The CVI measured 0.62 ± 0.04. The mean FAZ area was 0.48 ± 0.43 mm$^2$. The superficial and deep capillary plexus CDI was 0.66 ± 0.06 and 0.34 ± 0.08, respectively. AVR measured 0.67 ± 0.05.

## Follow-up image analyses among participants with obesity

Follow-up values of patients of Group A and B are listed in Tables 4 and 5. In Group A, the choroidal thickness showed a statistically significant increase at 3 months (329.27 ± 79; p<0.01). There was no significant change in the CVI among participants in Group A

**Table 3. Comparison of retinochoroidal vascular parameters between normal subjects and subjects with exogenous obesity.**

| Retinochoroidal parameters | Baseline | | | | p value* |
| --- | --- | --- | --- | --- | --- |
| | Normal Subjects (n = 30) | | Subjects with Obesity (n = 60) | | |
| | Mean | SD | Mean | SD | |
| RT (μm) | 279.5 | 28 | 269.23 | 15 | 0.14 |
| CT (μm) | 278.9 | 58 | 284.17 | 67 | 0.91 |
| CVI | 0.66 | 0.02 | 0.63 | 0.04 | **<0.01** |
| AVR | 0.7 | 0.04 | 0.68 | 0.05 | **<0.01** |
| Mean FAZ area (mm$^2$) | 0.26 | 0.08 | 0.45 | 0.32 | **<0.01** |
| CDI (superficial) | 0.7 | 0.04 | 0.68 | 0.06 | **0.04** |
| CDI (deep) | 0.38 | 0.02 | 0.35 | 0.06 | **0.01** |

* Mann-Whitney U Test.

AVR: arteriovenous ratio; CDI: capillary density index; CT: choroidal thickness; CVI: choroidal vascularity index; FAZ: foveal avascular zone; RT: retinal thickness; SD: standard deviation.

(0.61 ± 0.11; p = 0.19). The follow-up measurements on SS-OCTA did not reveal any significant change in the FAZ area in subjects from Group A (0.43 ± 0.16 mm$^2$; p = 0.99). Similarly, no change was noted in the CDI at 3 months in the superficial capillary plexus (0.69 ± 0.06; p = 0.6) or the deep capillary plexus (0.37 ± 0.05; p = 0.51). The AVR was higher at follow-up among subjects from Group A compared to baseline but not statistically significant (0.70 ± 0.05; p = 0.06).

Follow-up measurements in Group B at 3 months revealed no significant change in either the retinal or choroidal thickness compared to baseline (Table 5). Similar to the observations in Group A, there were no significant changes in the values of CVI (0.63 ± 0.02; p = 0.32), FAZ area (0.42 ± 0.13 mm$^2$; p = 0.73) and superficial capillary plexus CDI (0.69 ± 0.08; p = 0.06), and deep capillary plexus CDI (0.36 ± 0.05 p = 0.72). The AVR showed a significant increase compared to baseline values (0.69 ± 0.05; p = 0.03).

Multiple regression analysis was performed for the change in the choroidal thickness from baseline to 3 months based on the confounding variables of age, sex, systemic comorbidities, BMI, waist circumference, HbA1c, MAP, MOPP and the treatment received. The treatment

**Table 4. Baseline and follow-up values of retinochoroidal microvasculature among subjects with exogenous obesity undergoing bariatric surgery (Group A).**

| Retinochoroidal parameters | Baseline | | 3 months | | P-value* |
| --- | --- | --- | --- | --- | --- |
| | Mean | SD | Mean | SD | |
| Group A (n = 30) | | | | | |
| RT (μm) | 268.14 | 13 | 270.09 | 13 | 0.28 |
| CT (μm) | 272 | 73 | 329.27 | 79 | **<0.01** |
| CVI | 0.63 | 0.04 | 0.61 | 0.11 | 0.19 |
| Mean FAZ area (mm$^2$) | 0.43 | 0.13 | 0.43 | 0.16 | 0.99 |
| AVR | 0.68 | 0.06 | 0.70 | 0.05 | 0.06 |
| CDI (superficial) | 0.68 | 0.06 | 0.69 | 0.06 | 0.6 |
| CDI (deep) | 0.36 | 0.03 | 0.37 | 0.05 | 0.51 |

* Wilcoxon-Signed Rank Test.

AVR: arteriovenous ratio; CDI: capillary density index; CT: choroidal thickness; CVI: choroidal vascularity index; FAZ: foveal avascular zone; RT: retinal thickness; SD: standard deviation.

**Table 5. Baseline and follow-up values of retinochoroidal microvasculature among subjects with exogenous obesity receiving conservative management (Group B).**

| Retinochoroidal parameters | Baseline | | 3 months | | p value* |
|---|---|---|---|---|---|
| | Mean | SD | Mean | SD | |
| Group B (n = 30) | | | | | |
| RT (μm) | 269.99 | 16 | 271.09 | 16 | 0.32 |
| p value** | 0.74 | | 0.56 | | - |
| CT (μm) | 284.71 | 58 | 298.10 | 50 | 0.07 |
| p value** | 0.48 | | 0.82 | | - |
| CVI | 0.62 | 0.04 | 0.63 | 0.02 | 0.32 |
| p value** | 0.22 | | 0.31 | | - |
| Mean FAZ area (mm$^2$) | 0.48 | 0.43 | 0.42 | 0.13 | 0.73 |
| p value** | 0.88 | | 0.78 | | - |
| AVR | 0.67 | 0.05 | 0.69 | 0.05 | **0.03** |
| p value** | 0.82 | | 0.56 | | - |
| CDI (superficial) | 0.66 | 0.06 | 0.69 | 0.08 | 0.06 |
| p value** | 0.12 | | 0.96 | | - |
| CDI (deep) | 0.34 | 0.08 | 0.36 | 0.05 | 0.72 |
| p value** | 0.83 | | 0.54 | | |

* Wilcoxon-Signed Rank Test.

** Mann-Whitney U Test (comparison with subjects from Group A).

AVR: arteriovenous ratio; CDI: capillary density index; CT: choroidal thickness; CVI: choroidal vascularity index; FAZ: foveal avascular zone; RT: retinal thickness; SD: standard deviation.

effect was statistically significant with a p value of 0.021 after adjusting with the covariates. The mean choroidal thickness in Group A (surgery) was 61.11 μm higher than that of conservative treatment (Group B) after adjusting for the above covariates. The effect of all the covariates on the change in choroidal thickness was statistically insignificant (p > 0.05). A noticeable observation was, for unit increase in HbA1c, the mean change in choroidal thickness reduced by 12.146 μm (p = 0.07) although statistically insignificant.

## Between-group comparison

The mean retinal and choroidal thicknesses also did not differ statistically between the two groups. There was no significant difference between the two groups in comparing the baseline and follow-up CVI, FAZ and CDI (both superficial and deep capillary plexuses), and the AVR (Table 5).

## Discussion

Obesity results from morphological and functional changes in the adipose tissue, which is associated with changes in various inflammatory, hormonal, and metabolic factors [1,2,27]. Obesity can cause changes in the retinochoroidal microvasculature by a number of mechanisms, which may be the basis for ocular disease. Obesity is a multifactorial condition, and is associated with microvascular changes in several organ systems. Our study participants had a number of systemic comorbidities with obesity, demonstrating the multisystemic involvement in this condition. Increased adipose tissue in obesity has been associated with widening of venules, implicating microvascular dysfunction in the etiology of obesity [12]. Increased body weight and high BMI in obese patients can cause deranged metabolic profile due to the systemic oxidative stress secondary to hyperleptinemia [28]. Obese individuals have decreased

nitric oxide (NO) levels, which results in impaired dilatation of the vessels. Also, there are increased levels of certain vasoconstrictor molecules including endothelin-1 and angiotensin-II associated with higher BMI [3]. The choroidal blood flow is reduced through sympathetic activation and the release of noradrenalin, and increased through parasympathetic efferent nerve activation via NO signaling [29]. Lower levels of NO and increased levels of vasoconstrictor in obese patients may be responsible for decreased blood flow in the choroid.

In our study, we initially compared the retinochoroidal vascular parameters in patients with exogenous obesity with normal participants. Our study results demonstrated that patients with obesity have lower vascular retinal and choroidal vascular parameters such as CVI and CDI (Table 3). However, these participants do not have any significant difference in either the mean retinal or choroidal thickness compared to normal participants. The AVR was statistically higher in the normal subjects compared to patients with obesity (0.70 ± 0.03 versus 0.68 ± 0.05, respectively; p<0.01), which suggests that these subjects may have metabolic abnormalities including increased arteriolar resistance and systemic hypertension due to arteriolar narrowing. These vascular changes may affect the retinal circulation as well, demonstrated by altered AVR, resulting in reduced vascular flow of the retina and the optic nerve head. Reduced oxygenation and flow and may predispose these eyes to develop retinal vascular diseases. Thus, newer quantitative parameters on OCT and OCTA may provide important insights into the subclinical changes that occur in obese participants.

Previous studies have shown that the choroidal thickness is higher in patients with BMI between 18–24 compared to those with BMI < 18.5 (384 ± 102 μm versus 378 ± 86 μm; p<0.01) [18]. Similarly, studies in obese women and children have shown that choroidal thickness is higher in patients with obesity compared to healthy control population [30,31]. In our study, we also observed higher choroidal thickness values in participants with obesity compared to healthy controls (Table 3), though it did not reach statistical significance. Indices such as CVI and CDI measured using SS-OCTA may show pathological alterations along with choroidal thickness values in obese participants. SS-OCTA has not been used previously to study these ocular parameters in obese participants. In addition, our observations on AVR are in agreement with another previous study where the values were reported to be lower than normal control participants [12]. Thus, both retinal as well as choroidal microvasculature appears to be adversely affected in obesity.

Further, we evaluated the effect of weight loss by corrective intervention (bariatric surgery or diet/exercise) on various retinochoroidal parameters. Both the groups were benefitted by intervention with a statistically significant decrease in the mean waist circumference (in Group A) and BMI (in both the groups) (Table 2). In Group B, the increase in waist circumference could be explained by errors while obtaining manual measurements. The change in these parameters were significantly higher after bariatric surgery compared to conservative management with diet and exercise [32]. The mean choroidal thickness increased significantly from the baseline in patients undergoing bariatric surgery. In patients receiving conservative treatment, no significant change in choroidal thickness was noted at 3 months. The increase in the choroidal thickness could be attributed to the characteristic features of choroidal flow autoregulation and the impact of systemic parameters such as MAP. In our study, we observed that MAP significantly reduced in participants of Group A following bariatric surgery (not in Group B) (Table 2; p = 0.03). In the literature, studies have shown that the systemic MAP reduces both in the long-term and short-term in response to bariatric surgery [33,34]. Significant improvements occur in the hemodynamics and cardiovascular output in participants undergoing bariatric surgery. Choroidal blood flow is significantly affected by the systemic MAP. More recent studies using animal models in the past two decades have shown that significant alterations in choroidal blood flow can occur by manipulating MAP [35–38].

Therefore, an improvement in MAP may result in higher choroidal blood flow, leading to increased choroidal thickness.

While the choroidal thickness increased significantly in participants undergoing weight reduction surgery, there was no change in the vascular parameters such as CVI, CDI and FAZ area. Both groups, however, demonstrated an improvement in the AVR but the values were statistically significant only in Group B (conservative management with diet/exercise). Improved retinal vessel AVR can result from reversal of arteriolar constriction and improved venular caliber [13].

In summary, participants with obesity have reduced retinal and choroidal vascular indices compared to normal participants. Following bariatric surgery, an increase in the choroidal thickness values and AVR may occur due to altered hemodynamics. However, parameters such as CVI and CDI may not change despite bariatric surgery. Our study has numerous limitations including a modest sample size and relatively short follow-up. Certain confounding factors could have affected the analyses in our study. Majority of the participants with obesity had co-existing systemic diseases such as diabetes and systemic hypertension. However, the multivariate regression analysis did not reveal any significant effect of the confounding factors on choroidal thickness. The Hb1Ac values did however, show an effect on choroidal thickness but it was statistically insignificant. Only 2 participants in Group A, and 3 participants in Group B did not have any comorbidities. Since this is a small number, we could not perform any meaningful statistical subgroup analysis in participants with obesity without comorbidities. It would be ideal to include a higher sample size and participants with obesity and no vascular comorbidity. However, in the real world, since these diseases often co-exist, it is challenging to obtain such a cohort. We believe that our study also may suffer from selection bias. The prevalence of systemic comorbidities was higher in the subset of patients in Group A (those undergoing surgery) compared to Group B (managed conservatively), possibly due to higher BMI values and worse disease. However, this variability may not affect the overall comparison of patients (both groups combined) with normal participants, nor is it likely affect the before-after comparison of participants within the groups. In addition, normal control subjects were approximately 12.8 years younger than the patient population. This may represent a limitation in the analysis of the choroidal thickness, since its values are known to decrease by approximately 17 μm/decade after 30 years of age [17]. However, in a previous report, multiple regression model has shown that CVI is not statistically associated with age [20]. Obesity is typically a female preponderant disease. However, in India, gender disparities and other socio-economic issues may be factors that led to a male preponderance in our cohort [39,40]. In our study, certain parameters such as AVR may not have reached statistical significance due to smaller number of participants. However, our results are in sync with previous reports and further show the utility of quantitative measurement of retinochoroidal microvasculature.

In conclusion, our study shows that SS-OCT and SS-OCTA are very useful in demonstrating subclinical changes in the retinochoroidal microvasculature in systemic diseases such as obesity, and their partial reversal following corrective surgery/therapy.

## Supporting information

**S1 Data.**
(XLS)

## Acknowledgments

We would like to acknowledge efforts of Mr. Arun Kapil, Mr. Sushil Bhatt, and Mr. Nitin Gautam who helped in the acquisition of images in our case.

OCTA study group

1. Vishali Gupta (Lead of the consortium), Advanced Eye Center, Department of Ophthalmology, Post Graduate Institute of Medical Education and Research (PGIMER), Chandigarh, India. Email: vishalisara@yahoo.co.in

2. Aniruddha Agarwal, Advanced Eye Center, Department of Ophthalmology, Post Graduate Institute of Medical Education and Research (PGIMER), Chandigarh, India. Email: aniruddha9@gmail.com

3. Reema Bansal, Advanced Eye Center, Department of Ophthalmology, Post Graduate Institute of Medical Education and Research (PGIMER), Chandigarh, India. Email: drreemab@rediffmail.com

4. Rupesh Agrawal, National Healthcare Group Eye Institute, Department of Ophthalmology, Tan Tock Seng Hospital, Singapore. Email: rupeshttsh@gmail.com

5. Sarakshi Mahajan, School of Medicine, St Joseph Mercy Hospital, Oakland, Pontiac, Michigan (USA). Email: sarakshi424@gmail.com

6. Kanika Aggarwal, Advanced Eye Center, Department of Ophthalmology, Post Graduate Institute of Medical Education and Research (PGIMER), Chandigarh, India. Email: kanika2k1@yahoo.co.in

## Author Contributions

**Conceptualization:** Arshiya Saini, Sarakshi Mahajan, Rupesh Agrawal, Ashu Rastogi, Vishali Gupta.

**Data curation:** Aniruddha Agarwal, Sarakshi Mahajan, Rupesh Agrawal, Carol Y. Cheung, Ashu Rastogi, Rajesh Gupta, Yu Meng Wang, Michael Kwan, Vishali Gupta.

**Formal analysis:** Aniruddha Agarwal, Arshiya Saini, Sarakshi Mahajan, Rupesh Agrawal, Carol Y. Cheung, Ashu Rastogi, Rajesh Gupta, Yu Meng Wang, Michael Kwan, Vishali Gupta.

**Funding acquisition:** Arshiya Saini.

**Investigation:** Aniruddha Agarwal, Arshiya Saini, Vishali Gupta.

**Methodology:** Aniruddha Agarwal, Sarakshi Mahajan, Rupesh Agrawal, Carol Y. Cheung, Ashu Rastogi, Rajesh Gupta, Yu Meng Wang, Michael Kwan, Vishali Gupta.

**Project administration:** Aniruddha Agarwal, Arshiya Saini, Vishali Gupta.

**Resources:** Aniruddha Agarwal, Arshiya Saini, Vishali Gupta.

**Supervision:** Arshiya Saini, Rupesh Agrawal, Vishali Gupta.

**Validation:** Aniruddha Agarwal, Arshiya Saini, Sarakshi Mahajan, Rupesh Agrawal, Carol Y. Cheung, Ashu Rastogi, Rajesh Gupta, Yu Meng Wang, Michael Kwan, Vishali Gupta.

**Visualization:** Ashu Rastogi.

**Writing – original draft:** Aniruddha Agarwal, Arshiya Saini.

**Writing – review & editing:** Aniruddha Agarwal, Arshiya Saini, Sarakshi Mahajan, Rupesh Agrawal, Carol Y. Cheung, Ashu Rastogi, Rajesh Gupta, Yu Meng Wang, Michael Kwan, Vishali Gupta.

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
