## [Decision Letter · Decision Letter 0]

15 Apr 2020

PONE-D-20-08014

Effect of Weight Loss on the Retinochoroidal Structural Alterations Among Patients with Exogenous Obesity

PLOS ONE

Dear Dr. Gupta,

Thank you for submitting your manuscript to PLOS ONE. After careful consideration, we feel that it has merit but does not fully meet PLOS ONE’s publication criteria as it currently stands. Therefore, we invite you to submit a revised version of the manuscript that addresses the points raised during the review process.

We would appreciate receiving your revised manuscript by May 30 2020 11:59PM. To enhance the reproducibility of your results, we recommend that if applicable you deposit your laboratory protocols in protocols.io, where a protocol can be assigned its own identifier (DOI) such that it can be cited independently in the future. For instructions see: http://journals.plos.org/plosone/s/submission-guidelines#loc-laboratory-protocols

We look forward to receiving your revised manuscript.

Kind regards,

Pukhraj Rishi

Academic Editor

PLOS ONE

Journal Requirements:

 "The funders had no role in study design, data collection and analysis, decision to publish, or preparation of the manuscript"

3. One of the noted authors is a group or consortium 'OCTA Study Group'. In addition to naming the author group, please list the individual authors and affiliations within this group in the acknowledgments section of your manuscript. Please also indicate clearly a lead author for this group along with a contact email address.

4. Your ethics statement must appear in the Methods section of your manuscript. If your ethics statement is written in any section besides the Methods, please move it to the Methods section and delete it from any other section. Please also ensure that your ethics statement is included in your manuscript, as the ethics section of your online submission will not be published alongside your manuscript.

Additional Editor Comments (if provided):

The reviewers have raised several pertinent concerns including those related to the methodology of the study.

However, in view of the interesting research question, it is worthwhile to have the authors an opportunity to revise the manuscript.

Reviewers' comments:

Reviewer's Responses to Questions

**Comments to the Author**

1. Is the manuscript technically sound, and do the data support the conclusions?

Reviewer #1: Partly

Reviewer #2: No

Reviewer #3: Yes

2. Has the statistical analysis been performed appropriately and rigorously? 

Reviewer #1: No

Reviewer #2: Yes

Reviewer #3: No

3. Have the authors made all data underlying the findings in their manuscript fully available?

Reviewer #1: Yes

Reviewer #2: Yes

Reviewer #3: Yes

4. Is the manuscript presented in an intelligible fashion and written in standard English?

Reviewer #1: Yes

Reviewer #2: Yes

Reviewer #3: Yes

5. Review Comments to the Author

Reviewer #1: 1. Table 1 details baseline characteristics of Groups A and B but not the normal subjects.

2. Table 2 does not give the standard deviations or confidence intervals.

3. The authors defined the obesity as BMI>27.5 and waist circumference of more than 80cms. However, the authors do not give the actual distribution and comparison of the BMI and waist circumference between normals and obese subjects in this study. This may be important to make sure that the normal subjects are really normal.

4. The methodology does not specify where the choroidal thickness and retinal thickness were measured- in relation to fovea and whether the measurement taken for analysis is an average of several measurements?

5. One is not clear also whether the measurements within one subject were taken at roughly the same time of the day? There are studies to show significant diurnal variation in choroidal thickness of up to 33.7 micrometers (Tan CS, Ouyang Y, Ruiz H, Sadda SR. Diurnal variation of choroidal thickness in normal, healthy subjects measured by Spectral domain optical coherence tomography. IOVS 2012;53:261-266)

6. Are there any obese subjects without comorbidities? They would form a good subgroup for analysis of effect of obesity itself (if any) on the parameters studied.

7. To evaluate the role of coexisting vascular disorder Vs only obesity on the vascular parameters that have been studied, one would ideally need two controls- a) non-obese subjects with systemic vascular disorder (roughly of the same duration and severity as seen in the obese subjects. And b) as mentioned above- any subgroup of obese subjects without vascular comorbidity (perhaps such a group may be difficult to get).

8. The authors conclude –‘While the choroidal thickness increased significantly in subjects undergoing weight reduction surgery, there was no change in the vascular parameters such as CVI, CDI and FAZ area, indicating that these vascular alterations may be irreversible.’ There are several issues to this sweeping conclusion.

a) At baseline the authors found choroidal thickness and retinal thickness to be similar between normal subjects and obese subjects. But the obese subjects had increased choroidal thickness after bariatric surgery - does this mean their choroidal thickness is now supra normal?

b) The other vascular parameters- CVI, AVR, FAZ, CDI have not altered before and after. To conclude this- as indicative of permanent damage to vessels would also force one to admit that the choroidal thickness becomes supra normal after bariatric surgery!

c) The authors excluded all patients with obvious retinal vascular disorder among the subjects with obesity. Hence it is understandable that the systemic vascular disease is also likely to be of short duration and perhaps under adequate control.

Under such circumstances it would be difficult to accept a conclusion that attributes the lack of difference in the parameters before and after weight reduction to irreversible vascular changes.

Reviewer #2: The comparison of baseline features in subjects vs controls is skewed by significant prevalence of diabetes and hypertension, etc...which are known to affect choroidal vasculature and FAZ.

The groups A, B are different systemically- when systemic disease and treatment is being studied for its effect on an ocular finding, we expect the groups to be better matched.

Also, we do not know what happened to the systemic disease after bariatric surgery or wt reduction- did it affect blood pressures, DM control, etc that led to change in systemic medication?

We know that HTN/ DM get better with wt reduction and more so with bariatric surgery...so the systemic status of the groups and changes in medication after the intervention will be useful to know, and any meaningful conclusion can only be derived from that.

Where there are so many systemic variables confounding the basic disease (Obesity), the study would need a much larger sample size to have relevant statistics.

Reviewer #3: Comments

Obesity is a female preponderant disease. How do you explain male preponderance in your cases?

Was Diurnal variation accounted for while measuring choroidal thickness?

Co morbidities are mentioned at baseline but not at final follow up. What improvements were noted in co morbid factors like BP , Hb AIC at final follow up and their effect on final CT and other variables.?

CT depends on multitude of factors. Was multivariate analysis performed to look for the effect of other variables? Kindly provide the data

Review of literature reveals that CT is higher in patients with obesity. However the present study shows decreased CT in patients with obesity. This appears to be major conflicting result vis a vis published literature. Authors need to clarify this point in a more robust way in review of literature

( 1. Yilmaz I, Ozkaya A, Kocamaz M, et al. CORRELATION OF CHOROIDAL THICKNESS AND BODY MASS INDEX. Retina (Philadelphia, Pa). 2015;35(10):2085-2090. doi:10.1097/IAE.0000000000000582

2. Ophthalmic Surg Lasers Imaging Retina. 2017 Jan 1;48(1):10-17. doi: 10.3928/23258160-20161219-02. Choroidal Thickness in Childhood Obesity.

Bulus AD, Can ME, Baytaroglu A, Can GD, Cakmak HB, Andiran N.

3. BMC Ophthalmol. 2016 May 4;16(1):48. doi: 10.1186/s12886-016-0227-z.

Choroidal thickness in obese women. Yumusak E1, Ornek K2, Durmaz SA3, Cifci A4, Guler HA2, Bacanli Z4.

6. PLOS authors have the option to publish the peer review history of their article (what does this mean?). If published, this will include your full peer review and any attached files.

Reviewer #1: No

Reviewer #2: No

Reviewer #3: No

---

## [Author Response · Author response to Decision Letter 0]

18 May 2020

May 11, 2020

Pukhraj Rishi, MD 

Academic Editor

PLOS ONE

RE: PONE-D-20-08014: Effect of Weight Loss on the Retinochoroidal Structural Alterations Among Patients with Exogenous Obesity 

Dear Dr. Rishi:

On behalf of all the authors, I wish to thank you and the reviewers helping us improve our manuscript. We appreciate the comments and have addressed them as indicated below. 

Journal Requirements:

Response 1: Thank you very much for providing the links to the PLOS ONE style templates. We have incorporated these in the revised manuscript. 

"The funders had no role in study design, data collection and analysis, decision to publish, or preparation of the manuscript"

Response 2: We have noted the above instructions. We have not received any funding for this work. The statement, “The authors received no specific funding for this work” has been added to the Cover Letter. Thank you. 

3. One of the noted authors is a group or consortium 'OCTA Study Group'. In addition to naming the author group, please list the individual authors and affiliations within this group in the acknowledgments section of your manuscript. Please also indicate clearly a lead author for this group along with a contact email address.

Response 3: Thank you very much for the comment. As per the journal requirements, we have stated the details of the OCTA study group in the acknowledgements section of the manuscript. Individual email addresses are also listed. 

4. Your ethics statement must appear in the Methods section of your manuscript. If your ethics statement is written in any section besides the Methods, please move it to the Methods section and delete it from any other section. Please also ensure that your ethics statement is included in your manuscript, as the ethics section of your online submission will not be published alongside your manuscript.

Response 4: Thank you very much for the comment. We have added the following statement in the methods section, “The study was approved by the Institute Ethics Committee of PGIMER and adhered to the tenets of Declaration of Helsinki.”

Page 8, First Paragraph

Additional Editor Comments:

The reviewers have raised several pertinent concerns including those related to the methodology of the study. However, in view of the interesting research question, it is worthwhile to have the authors an opportunity to revise the manuscript.

Response: We would like to thank the Additional Editor for thoroughly reviewing our manuscript, and giving us the chance to respond to the reviewer’s comments. We have significantly modified the manuscript based on the comments, clarified several aspects of the methodology, and improved reporting of the results. We hope that the revisions are appropriate, and the manuscript can be considered further. 

Reviewer #1: 

1. Table 1 details baseline characteristics of Groups A and B but not the normal subjects.

Response 1: Thank you very much for the comment. We have included baseline characters of normal control population in Table 1. 

2. Table 2 does not give the standard deviations or confidence intervals.

Response 2: Thank you very much for the suggestions. We have included the standard deviation values in the Table 2. 

3. The authors defined the obesity as BMI>27.5 and waist circumference of more than 80 cm. However, the authors do not give the actual distribution and comparison of the BMI and waist circumference between normal and obese subjects in this study. This may be important to make sure that the normal subjects are really normal.

Response 3: Thank you very much for highlighting this important aspect. We have mentioned the height, weight, BMI and waist circumference of the normal control subjects included in this study in the results section. The normal subjects had a mean BMI of 24.90 and mean waist circumference of 76.07 cm.

Results section; Page 13, 1st paragraph 

4. The methodology does not specify where the choroidal thickness and retinal thickness were measured - in relation to fovea and whether the measurement taken for analysis is an average of several measurements?

Response 4: Thank you once again for highlighting this pertinent aspect. We have added the missing information in the methods. The measurements were performed at the fovea. The following statement has been added, “The measurements were performed at the fovea by two independent graders. The average of the two graders’ measurements was used for analysis.”

Methods section; Page 10; 2nd paragraph

5. One is not clear also whether the measurements within one subject were taken at roughly the same time of the day? There are studies to show significant diurnal variation in choroidal thickness of up to 33.7 micrometers (Tan CS, Ouyang Y, Ruiz H, Sadda SR. Diurnal variation of choroidal thickness in normal, healthy subjects measured by Spectral domain optical coherence tomography. IOVS 2012;53:261-266)

Response 5: As the reviewer suggested, it is important to keep in mind the diurnal variation of the choroidal thickness and therefore, all measurements need to be performed at a particular time interval in the day. In our study, all the measurements were performed between 10.00 am to 12.00 noon. This is also the common working hours of the Obesity Clinic and Ophthalmology Clinic in our institute. We have mentioned this information in the revised manuscript.

Page 9; Section on “Acquisition of Images” 

6. Are there any obese subjects without comorbidities? They would form a good subgroup for analysis of effect of obesity itself (if any) on the parameters studied.

Response 6: In our cohort, among the patients in Group A (those undergoing surgery), there were only 2 patients who did not have additional co-morbidities. Likewise, in Group B (those receiving conservative therapy), 3 patients did not have any additional co-morbidities. Therefore, we believe that this is a small number that may not allow meaningful statistical analysis. Unfortunately, subjects with obesity tend to have various comorbidities, and this has been mentioned in the limitations of the manuscript.

Page 19, Last line; Page 20, first 3 lines

7. To evaluate the role of coexisting vascular disorder versus only obesity on the vascular parameters that have been studied, one would ideally need two controls- a) non-obese subjects with systemic vascular disorder (roughly of the same duration and severity as seen in the obese subjects. And b) as mentioned above- any subgroup of obese subjects without vascular comorbidity (perhaps such a group may be difficult to get).

Response 7: Thank you very much for the insightful comment. We agree with the reviewer that it would be ideal to have two such groups – one of non-obese subjects with systemic vascular diseases of same duration and severity, and obese subjects without vascular comorbidity. But as the reviewer suggests, in the real world, it is very difficult to obtain such a cohort especially because some of these diseases co-exist. We have added this factor in the discussion, highlighting the limitations of our selection.

Page 20; First paragraph 

8. The authors conclude – ‘While the choroidal thickness increased significantly in subjects undergoing weight reduction surgery, there was no change in the vascular parameters such as CVI, CDI and FAZ area, indicating that these vascular alterations may be irreversible.’ There are several issues to this sweeping conclusion.

a) At baseline the authors found choroidal thickness and retinal thickness to be similar between normal subjects and obese subjects. But the obese subjects had increased choroidal thickness after bariatric surgery - does this mean their choroidal thickness is now supra normal?

b) The other vascular parameters - CVI, AVR, FAZ, CDI have not altered before and after. To conclude this- as indicative of permanent damage to vessels would also force one to admit that the choroidal thickness becomes supra normal after bariatric surgery!

c) The authors excluded all patients with obvious retinal vascular disorder among the subjects with obesity. Hence it is understandable that the systemic vascular disease is also likely to be of short duration and perhaps under adequate control.

Under such circumstances it would be difficult to accept a conclusion that attributes the lack of difference in the parameters before and after weight reduction to irreversible vascular changes.

Response 8: Thank you very much for the comments. We agree with your observations and accordingly, have performed additional literature search to understand the choroidal flow and autoregulation in systemic diseases. In response to point b, we also calculated Mean ocular perfusion pressure (MOPP) as one of the vascular flow regulatory factor in the ocular circulation and possibly affecting CVI, AVR and CDI. The revised manuscript now provides an explanation as to why the choroidal thickness could have increased following surgery. 

We observed an interesting finding in our study. The mean arterial pressure (MAP) significantly reduced in Group A (those subjects undergoing surgery) and not in Group B (patients managed conservatively). However, there was no statistically significant difference in MOPP in both the groups. This has been added to the revised Table 2. Studies have shown that the systemic MAP reduces both in the long-term and short-term in response to bariatric surgery. Choroidal blood flow is significantly affected by MAP, and therefore, it may result in increased choroidal thickness postoperatively in patients of Group A. Hence, the choroidal thickness should not be labeled as ‘supra normal’. 

Among the other vascular parameters, CVI and CDI were lower in obese subjects compared to normal healthy controls. However, we have removed the argument that change in CVI and CDI indicates permanent damage to these structures.

Page 19, multiple places 

Reviewer #2: 

1. The comparison of baseline features in subjects versus controls is skewed by significant prevalence of diabetes and hypertension, etc. which are known to affect choroidal vasculature and FAZ. The groups A, B are different systemically - when systemic disease and treatment is being studied for its effect on an ocular finding, we expect the groups to be better matched.

Response 1: We would like to thank the reviewer for the insightful comments. The baseline comorbidities are different among subjects in Groups A and B. This is likely due to a selection bias, i.e. subjects who are in Group A and undergoing bariatric surgery may be the ones with higher BMI (as reflected in Table 2), and with higher BMI, complications and comorbidities such as hypertension and sleep apnea can be higher than those in Group B (patients not undergoing surgery). Therefore, this has been mentioned in the revised manuscript in the limitations.

However, the variability of the comorbidities and systemic disease between Group A and Group B will not affect the overall comparison of patients (both groups combined) with normal subjects, nor will it affect the before-after comparison of subjects within the groups. Therefore, we believe that our data set is still pertinent and with the inherent limitations, comparison between the groups is still possible. Thank you very much.

Page 20; Discussion 

2. Also, we do not know what happened to the systemic disease after bariatric surgery or weight reduction - did it affect blood pressures, DM control, etc?

Response 2: Thank you very much for the comment. We have already reported the follow-up changes in the weight, BMI, waist circumference, and HB1AC in patients from Group A and B in Table 2. Additionally, we have added the mean values of systolic and diastolic blood pressures in the two groups at baseline and follow-up in Table 2. We also calculated and added the systemic mean arterial pressure (MAP) in Table 2, and realized that subjects undergoing bariatric surgery (Group A) had a significantly reduced MAP in the follow-up period. 

In the follow-up, the subjects also experienced a statistically significant improvement in parameters such as weight (in Group A), BMI, waist circumference and HB1AC. However, there was no statistically significant difference in the mean systolic and diastolic blood pressure. This information is provided in Table 2. 

3. We know that HTN/ DM get better with weight reduction and more so with bariatric surgery... so the systemic status of the groups and changes in medication after the intervention will be useful to know, and any meaningful conclusion can only be derived from that.

Response 3: Thank you very much for the comment. In Group A (the subjects who underwent bariatric surgery), the systolic and diastolic blood pressure improved at follow-up, but did not reach statistical significance. However, there was improvement in diabetes (reflected by HB1AC values). Thank you very much for the suggestion.

Results; Table 2 

4. Where there are so many systemic variables confounding the basic disease (Obesity), the study would need a much larger sample size to have relevant statistics.

Response 4: The sample size has been calculated for the study based on the study by Dogan et al [26] considering equal allocation ratio of 1:1, which will provide the desired effect with 95% confidence and 80% power. As per Dogan et al, the choroidal thickness for obese group was 300.63 ± 65.55 µm, while that of normal group was 338.79 ± 64.41 µm. Accordingly, the effect size obtained was 0.59 (~0.60). In the current study, a sample size of in each group was obtained as 30. However, we do agree that a larger data set would help us understand the changes in the parameters better. Large dataset is challenging to obtain, since the number of patients undergoing bariatric surgery may be less. Hence, in the discussion, we have stated that one of the limitations of our study is the modest sample size.

Methods Section; Page 12; 2nd paragraph 

Reviewer #3: 

1. Obesity is a female preponderant disease. How do you explain male preponderance in your cases? Was diurnal variation accounted for while measuring choroidal thickness? Co-morbidities are mentioned at baseline but not at final follow up. What improvements were noted in co-morbid factors like BP, Hb AIC at final follow up and their effect on final CT and other variables?

Response 1: Thank you very much for highlighting these pertinent aspects. We agree that obesity is a female preponderant disease. However, in India, gender disparities in health care expenditures are well known due to various socioeconomic factors. Studies have shown that the health care expenditures on adult Indian females is systemically lower than that of adult males. Such socioeconomic disparities in health and health care are major concerns in India. We have added these references and the possible selection bias in the limitations. 

We performed the OCT scans between 10.00 am to 12.00 noon in our study to minimize the effect of diurnal variations (explained in the response to reviewer #1, comment 5).

We have now mentioned the comorbidities at follow-up as well. The improvements in HB1AC, BP and mean arterial pressure has been added to the revised manuscript. 

Page 20; Discussion; Table 2; multiple places 

2. CT depends on multitude of factors. Was multivariate analysis performed to look for the effect of other variables? Kindly provide the data

Response 2: Thank you very much for raising this very valid point. We have also added Mean ocular perfusion pressure (MOPP) in addition to all the possible factors which can affect CT and other vascular parameters in the eye. We have included the multiple regression analysis in the revised results. The multiple regression reveals that none of the confounders had a significant impact on the change in choroidal thickness from baseline to 3 months. It is noteworthy that with every unit increase in HB1AC, there was a change in the choroidal thickness of approximately 12 microns, but this was not statistically significant. A limitation, therefore, has been added to the discussion that a higher sample would be better for a detailed analysis.

An interesting aspect was that the only variable associated with higher choroidal thickness on multivariate analysis was the treatment strategy (significantly higher in those undergoing bariatric surgery). 

Page 15; paragraph 3

3. Review of literature reveals that CT is higher in patients with obesity. However the present study shows decreased CT in patients with obesity. This appears to be major conflicting result vis a vis published literature. Authors need to clarify this point in a more robust way in review of literature

1. Yilmaz I, Ozkaya A, Kocamaz M, et al. CORRELATION OF CHOROIDAL THICKNESS AND BODY MASS INDEX. Retina (Philadelphia, Pa). 2015;35(10):2085-2090. doi:10.1097/IAE.0000000000000582

2. Ophthalmic Surg Lasers Imaging Retina. 2017 Jan 1;48(1):10-17. doi: 10.3928/23258160-20161219-02. Choroidal Thickness in Childhood Obesity.

Bulus AD, Can ME, Baytaroglu A, Can GD, Cakmak HB, Andiran N.

3. BMC Ophthalmol. 2016 May 4;16(1):48. doi: 10.1186/s12886-016-0227-z.

Choroidal thickness in obese women. Yumusak E1, Ornek K2, Durmaz SA3, Cifci A4, Guler HA2, Bacanli Z4.

Response 3: Thank you very much for the comment. We agree that the previous studies listed by the reviewer have reported a higher choroidal thickness in subjects with obesity compared to healthy controls. In our study, the choroidal thickness in normal subjects was 278.9 microns, and 284.2 microns in subjects with obesity. Thus, we also found higher choroidal thickness values compared to normal subjects, but this did not reach statistical significance. 

The manuscript by Yilmaz et al showed that subjects with BMI between 18.5-24.9 have higher choroidal thickness than those with BMI > 25. In the study by Bulus et al, the choroidal thickness was higher in obese children. Yumusak et al have also shown higher choroidal thickness values in obese women compared to control population. The revised statements and newer references have been added in the revised manuscript. 

Page 17; Paragraph 2

Author Changes

We have added the values for mean ocular perfusion pressure (MOPP) calculated from mean arterial pressure and IOP. This has been mentioned in the methods (Page 9; first paragraph) and results (Page 13; 2nd paragraph). 

We hope our responses will satisfy the reviewers. We thank the Editor and the reviewers for dedicating their time and effort to review our manuscript. We hope that the manuscript is now appropriate for publication in Retina.

Sincerely,

Vishali Gupta, MS for the authors

---

## [Decision Letter · Decision Letter 1]

18 Jun 2020

PONE-D-20-08014R1

Effect of Weight Loss on the Retinochoroidal Structural Alterations Among Patients with Exogenous Obesity

PLOS ONE

Dear Dr. Gupta,

Thank you for submitting your manuscript to PLOS ONE. After careful consideration, we feel that it has merit but does not fully meet PLOS ONE’s publication criteria as it currently stands. Therefore, we invite you to submit a revised version of the manuscript that addresses the points raised during the review process.

Authors have done well to address the reviewers' concerns. However, there are some aspects that need attention and are detailed below.

We look forward to receiving your revised manuscript.

Kind regards,

Pukhraj Rishi

Academic Editor

PLOS ONE

Additional Editor Comments (if provided):

Authors have done well to address some major concerns of the reviewers. However, few clarifications need attention.

Reviewers' comments:

Reviewer's Responses to Questions

**Comments to the Author**

1. If the authors have adequately addressed your comments raised in a previous round of review and you feel that this manuscript is now acceptable for publication, you may indicate that here to bypass the “Comments to the Author” section, enter your conflict of interest statement in the “Confidential to Editor” section, and submit your "Accept" recommendation.

Reviewer #1: All comments have been addressed

Reviewer #3: (No Response)

2. Is the manuscript technically sound, and do the data support the conclusions?

Reviewer #1: Yes

Reviewer #3: Yes

3. Has the statistical analysis been performed appropriately and rigorously? 

Reviewer #1: Yes

Reviewer #3: Yes

4. Have the authors made all data underlying the findings in their manuscript fully available?

Reviewer #1: Yes

Reviewer #3: Yes

5. Is the manuscript presented in an intelligible fashion and written in standard English?

Reviewer #1: Yes

Reviewer #3: Yes

6. Review Comments to the Author

Reviewer #1: (No Response)

Reviewer #3: (No Response)

7. PLOS authors have the option to publish the peer review history of their article (what does this mean?). If published, this will include your full peer review and any attached files.

Reviewer #1: No

Reviewer #3: No

---

## [Author Response · Author response to Decision Letter 1]

20 Jun 2020

June 20, 2020

Pukhraj Rishi, MD 

Academic Editor

PLOS ONE

RE: PONE-D-20-08014R1: Effect of Weight Loss on the Retinochoroidal Structural Alterations Among Patients with Exogenous Obesity 

Dear Dr. Rishi:

On behalf of all the authors, I wish to thank you and the reviewers helping us improve our manuscript. We appreciate the comments and have addressed them as indicated below. 

Additional Editor Comments:

1. Authors have quoted a study of normative data based on imaging of 81 eyes with SDOCT, while their own study was performed with SSOCT. Suggest reference to another study on normative data based on imaging of 230 eyes with SSOCT may be more appropriate. Akhtar Z, Rishi P, Srikanth R, Rishi E, Bhende M, Raman R. Choroidal thickness in normal Indian subjects using Swept source optical coherence tomography. PLoS One. 2018;13(5):e0197457. Published 2018 May 16. doi:10.1371/journal.pone.0197457

Response: Thank you very much for the suggestion. We have added the suggested reference to the manuscript. 

New reference 17

2. If all the patients were recruited from PGI Chandigarh then how is HIPAA relevant in India?

Response: This statement has been removed from the manuscript. 

Page 8, 1st Paragraph, Last line

3. Rephrase this sentence (The composite third area was added to the to the ROI manager. 

Response: The statement has been corrected. Thank you. 

Page 11, 1st Paragraph, 3rd Line

4. The title of the Table represents data from group B but seems to indicate the cohort underwent bariatric surgery!

Response: We apologize for the error. The table 5 represents data from Group B, i.e. patients receiving conservative management. 

Table 5

5. “Our study supports the hypothesis that disturbance in the balance between systemic vasodilator and vasoconstrictor levels affects the choroidal blood flow.”

How? This is a broad statement and the study is not designed to answer this question. You might be asked to substantiate with biochemical parameters.

Response: We agree with the reviewer, and this statement has been removed from the manuscript. 

Page 17, 1st Paragraph, Last 3 Lines

6. “Both the groups were benefitted by intervention with a statistically significant decrease in the mean waist circumference and BMI.”

On the contrary the waist circumference increased in group B (Table 2). Please revise the statement, and how do you explain the difference in the two groups.

Response: Thank you very much for the comment. We have revised the statement as “Both the groups were benefitted by intervention with a statistically significant decrease in the mean waist circumference (in Group A) and BMI (in both the groups) (Table 2).” One explanation of this phenomenon could be an error in the measurements of the waist. 

Page 18, 2nd Paragraph, Lines 4-5

7. It was interesting to note that in group A, post-surgery, CVI decreased although not in a statistically significant way. So, we need to continue to follow-up these patients for a better understanding.

Response: We agree with the reviewer. We shall continue long-term follow-up of these patients. Thank you very much. 

8. Mean age was much younger than Obesity groups. Any comments?

Response: Thank you very much for the comment. The mean age difference between controls and subjects in our study was approximately 12.8 years. Based on the previous publications, the choroidal thickness is reduced by approximately 17 µm per decade after 30 years of age (Akhtar et al). This limitation has been provided in the manuscript. However, Agrawal et al have shown that on multiple regression model, CVI is not statistically associated with age. This has been added to the revised manuscript. 

Page 20, 1st Paragraph.

We hope our responses will satisfy the reviewers. We thank the Editor and the reviewers for dedicating their time and effort to review our manuscript. We hope that the manuscript is now appropriate for publication in Retina.

Sincerely,

Vishali Gupta, MS for the authors

---

## [Editor Report · Decision Letter 2]

25 Jun 2020

Effect of Weight Loss on the Retinochoroidal Structural Alterations Among Patients with Exogenous Obesity

PONE-D-20-08014R2

Dear Dr. Gupta,

We’re pleased to inform you that your manuscript has been judged scientifically suitable for publication and will be formally accepted for publication once it meets all outstanding technical requirements.

Kind regards,

Pukhraj Rishi

Academic Editor

PLOS ONE

Additional Editor Comments:

Authors have responded satisfactorily to all the concerns raised. Manuscript is now acceptable.
---

## [Editor Report · Acceptance letter]

26 Jun 2020

PONE-D-20-08014R2 

Effect of Weight Loss on the Retinochoroidal Structural Alterations Among Patients with Exogenous Obesity 

Dear Dr. Gupta:

I'm pleased to inform you that your manuscript has been deemed suitable for publication in PLOS ONE. Congratulations! Your manuscript is now with our production department. 

Kind regards, 

on behalf of

Dr. Pukhraj Rishi 

Academic Editor

PLOS ONE